

# Lagrangian characterization of heat waves: The perspective matters

Amelie Mayer[1] and Volkmar Wirth[1]

[1]Institute for Atmospheric Physics, Johannes Gutenberg University Mainz, Becherweg 21, 55126 Mainz, Germany

**Correspondence:** Amelie Mayer (amelie.mayer@uni-mainz.de)

**Abstract.** Although heat waves are one of the most dangerous types of weather-related hazards, their underlying mechanisms are not yet sufficiently understood. Especially, there is still no scientific consensus about the relative importance of the three key processes: horizontal temperature transport, subsidence accompanied by adiabatic heating, and diabatic heating. The current study quantifies these processes using a Eulerian method based on tracer advection, which allows one to extract Lagrangian information. For each grid point at any time, the method yields a decomposition of temperature anomalies into the aforementioned processes, complemented by the contribution of a pre-existing anomaly. Two different approaches for this decomposition are employed. The first approach is based on full (absolute) fields of the respective terms, whereas the second approach is based on anomaly fields of the respective terms, i.e., deviations from their corresponding climatologies. The two approaches offer two distinct perspectives on the same subject matter. By analyzing two recent heat waves, it is shown that the two decompositions yield substantial differences regarding the relative importance of the processes. A statistical analysis indicates that these differences are not coincidental, but characteristic for the respective regions. It is concluded that the Lagrangian characterization of heat waves is a matter of perspective.

## 1 Introduction

Heat waves stand out as one of the most perilous weather-related hazards. They threaten natural ecosystems and society in many ways (e.g. Perkins, 2015; Horton et al., 2016; Barriopedro et al., 2023, and references therin). Their frequency and intensity is projected to increase in the course of global warming (e.g. IPCC, 2021), likely posing an even greater danger to humanity in the future. The socioeconomic importance of heat waves is thus abundantly clear.

Far less clear, however, are the mechanisms that determine the formation of heat waves. While it is well known that heat waves in the extratropics are invariably associated with large-scale upper-tropospheric ridging (Sousa et al., 2018) or anticyclonic flow (e.g. Stefanon et al., 2012; Zschenderlein et al., 2019), often in the form of atmospheric blocking (e.g. Pfahl and Wernli, 2012), there remains a lack of consensus about how exactly air masses with anomalously high temperatures form in this large-scale setting.

Three processes are believed to play a role in the context of heat waves: horizontal advection of warm air (e.g. Screen, 2014; Harpaz et al., 2014; Garfinkel and Harnik, 2017; Sousa et al., 2019), adiabatic warming in subsiding air (e.g. Bieli et al., 2015; Zschenderlein et al., 2019), and diabatic heating near the surface (e.g. Miralles et al., 2014; Schumacher et al., 2019). The large-scale setting seems to be conducive to all three processes, and often more than one process has been considered to be



important (e.g. Quinting and Reeder, 2017; Hochman et al., 2021; Röthlisberger and Papritz, 2023). The question still under debate is to what extent each of the individual processes contributes to the formation of heat extremes.

A novel quantitative approach to address this question has recently been proposed by Röthlisberger and Papritz (2023). This approach forsees the decomposition of a temperature anomaly at a given location into (distinct) contributions from each of the three individual processes. More precisely, the approach quantifies the effect of horizontal advection across climatological temperature gradients, the combined effect of vertical advection across climatological temperature gradients and adiabatic warming, and the effect of parcel-based diabatic heating. Like many studies focussing on the processes within heat waves (Harpaz et al., 2014; Bieli et al., 2015; Zschenderlein et al., 2019; Schielicke and Pfahl, 2022), the temperature anomaly decomposition approach by Röthlisberger and Papritz (2023) adopts the Lagrangian framework. This is a natural framework for studying the processes under consideration, since the laws of dynamics and thermodynamics inherently apply to air parcels. In the Lagrangian framework, individual fluid parcels can be tracked, their physical properties can be identified, and any temporal changes in these properties can be assessed. Therefore, the Lagrangian framework enables a physically meaningful quantification of desired quantities such as adiabatic and (parcel-based) diabatic heating.

Given that most common atmospheric models work in the Eulerian framework, obtaining Lagrangian information about the flow requires some extra effort. Often, backward trajectory models are employed for that purpose (e.g., LAGRANTO, see Sprenger and Wernli 2015; HYSPLIT, see Stein et al. 2015). In this study, however, we refrain from calculating trajectories and instead opt for an alternative approach. We will use the method of tracer advection proposed in Mayer and Wirth (2023) to extract the necessary Lagrangian information. The strength of this method is that it provides the required Lagrangian information directly in the form of gridded fields available at any time step. Most importantly, this makes it quite straightforward to calculate climatologies. We will take advantage of this and, for the first time, present climatologies of the contributions from horizontal transport, vertical transport, and diabatic heating. Based on these climatologies, we will show that the original temperature anomaly decomposition proposed by Röthlisberger and Papritz (2023) may not be as unique as it may appear at first sight. We will, thus, offer a fresh perspective on the relevance of the individual processes for the formation of heat waves.

In this paper, we perform a temperature anomaly decomposition (Section 2.1), akin to the method proposed by Röthlisberger and Papritz (2023). To this end, we apply a novel technique to extract the required Lagrangian information (Section 2.2). We then use this setup to analyse two recent heat waves (Section 3.1) and to relate the contributions from the individual processes to their long-term averages (Section 3.2). Based on these long-term averages, we suggest an alternative decomposition based on anomaly fields (Section 3.3). Finally, we complement the two case studies by a short statistical analysis (Section 3.4). We close the paper with a brief discussion (Section 4) and a concluding summary (Section 5).

## 2 Method and Data

### 2.1 Lagrangian $\theta'$ decomposition

In this study, we aim to quantify the extent to which horizontal transport, vertical transport and diabatic heating contribute to the formation of temperature anomalies. For this purpose, we examine anomalies in potential temperature $\theta'$, which we decompose





from a Lagrangian perspective into the aforementioned processes. We opt for potential temperature instead of temperature as our metric because potential temperature is materially conserved in adiabatic flow, rendering it more convenient for theoretical considerations.

We consider the potential temperature anomaly $\theta'(\boldsymbol{x}, t)$ of an air parcel located at grid point $\boldsymbol{x}$ at time $t$. This anomaly, $\theta'(\boldsymbol{x}, t)$, represents the deviation of the potential temperature $\theta(\boldsymbol{x}, t)$ from its climatological average $\overline{\theta}(\boldsymbol{x}, t)$:

$$\theta'(\boldsymbol{x}, t) = \theta(\boldsymbol{x}, t) - \overline{\theta}(\boldsymbol{x}, t). \tag{1}$$

Taking the material derivative $D/Dt$ on both sides

$$\frac{D\theta'}{Dt} = \frac{D\theta}{Dt} - \frac{D\overline{\theta}}{Dt} \tag{2}$$

and applying Euler's relation (in pressure coordinates)

$$\frac{D}{Dt} = \frac{\partial}{\partial t} + \boldsymbol{v} \cdot \nabla_h + \omega \cdot \frac{\partial}{\partial p} \tag{3}$$

to the last term on the right hand side of (2), one obtains

$$\frac{D\theta'}{Dt} = \frac{D\theta}{Dt} - \frac{\partial \overline{\theta}}{\partial t} - \boldsymbol{v} \cdot \nabla_h \overline{\theta} - \omega \frac{\partial \overline{\theta}}{\partial p} \, , \tag{4}$$

where $\boldsymbol{v}$ denotes the horizontal wind, $\boldsymbol{\nabla_h}$ is the horizontal gradient, $p$ is pressure, and $\omega$ is the vertical wind in pressure coordinates. Finally we accumulate the contributions from each of the individual terms in (4) along the trajectory of the air parcel and obtain

$$\theta'(x, t) = -\int_{t_0}^{t} \boldsymbol{v} \cdot \boldsymbol{\nabla_h} \overline{\theta} dt' - \int_{t_0}^{t} \frac{\partial \overline{\theta}}{\partial p} \omega dt' + \int_{t_0}^{t} \frac{D\theta}{Dt'} dt' - \int_{t_0}^{t} \frac{\partial \overline{\theta}}{\partial t'} dt' + \theta_0' \, , \tag{5}$$

where $t_0$ is the start time of the integration, $t$ is the time of interest, and $\theta_0'$ is the temperature anomaly at start time. The integrals in (5) represent Lagrangian integrals, i.e., they accumulate information along the trajectory of the air parcel that happens to be located at location $\boldsymbol{x}$ at time $t$.

Equation (5) decomposes a potential temperature anomaly of an air parcel into five contributions: (1) horizontal transport across climatological horizontal temperature gradients; (2) vertical transport across climatological vertical temperature gradients; (3) diabatic heating of the air parcel along its trajectory; (4) local changes of the climatological potential temperature including seasonality and the diurnal cycle; and (5) the initial temperature anomaly of the air parcel at the start time of the integration. The above decomposition closely follows the approach by Röthlisberger and Papritz (2023), except that they used temperature instead of potential temperature and chose the initial time $t_0$ such that the temperature anomaly is zero at that time.

The integrals on the right hand side of (5) can be extended to start at $-\infty$ if, at the same time, the integrand is multiplied by a Heaviside function that jumps from 0 to 1 at time $t_0$. For reasons that will become clear later, we prefer to use an exponential function instead of a Heaviside function. As shown in Appendix A, this leads to the following equivalent formulation for



$\theta'(\boldsymbol{x}, t)$, namely

$$\theta'(\boldsymbol{x}, t) = -\int_{-\infty}^{t} \boldsymbol{v} \cdot \boldsymbol{\nabla_h} \overline{\theta} \ e^{-\lambda(t-t')} dt' - \int_{-\infty}^{t} \frac{\partial \overline{\theta}}{\partial p} \omega \ e^{-\lambda(t-t')} dt' + \int_{-\infty}^{t} \frac{D\theta}{Dt'} \ e^{-\lambda(t-t')} dt' - \int_{-\infty}^{t} \frac{\partial \overline{\theta}}{\partial t'} \ e^{-\lambda(t-t')} dt' + R \qquad (6)$$

with

$$R = \int_{-\infty}^{t} \left( -\boldsymbol{v} \cdot \boldsymbol{\nabla_h} - \frac{\partial \overline{\theta}}{\partial p} \omega + \frac{D\theta}{dt} + \frac{\partial \overline{\theta}}{\partial t'} \right) \left( 1 - e^{-\lambda(t-t')} \right) dt' \ .$$

Effectively, the contributions from the different processes in the first four terms of (6) are multiplied by a weighting factor $W = \exp[-\lambda(t-t')]$ with $\lambda > 0$; thus, $W$ decreases exponentially as the integration variable $t'$ goes backward in time. On the other hand, the different processes occurring in the $R$-term are multiplied by $1 - W$ such that $R$ essentially contains contributions that occurred in the more distant past. The term $R$ in (6) thus corresponds to $\theta'_0$ in (5), although they are not identical. The constant $\lambda > 0$ is chosen such that the associated time scale $\lambda^{-1}$ is of the same order of magnitude as $t - t_0$. Like in (5), the integrals in the above equation represent accumulation of processes on air parcels along their trajectories. The only difference is that now the accumulation is more gradually spread over time such that the contributions fade away more gradually rather than suddenly at time $t_0$. The decomposition in (6),

$$\theta' = \theta_{hor} + \theta_{ver} + \theta_{dia} + \theta_{sea} + \theta_{pre} \ , \qquad (7)$$

is the basis for the analysis in our paper. In the following, we are going to refer to the first three terms as the "process terms", comprising the contributions from (1) horizontal transport $\theta_{hor}$, (2) vertical transport $\theta_{ver}$, and (3) diabatic heating $\theta_{dia}$ to the temperature anomaly $\theta'$. The fourth term, $\theta_{sea}$, containing the seasonality is much smaller than all other terms and will be neglected in the remainder of this paper. The last term, $\theta_{pre}$, corresponds to $R$ and is interpreted as the pre-existing potential temperature anomaly.

### 2.1.1 Eulerian tracer advection with relaxation

One way to obtain the Lagrangian information required to evaluate the Lagrangian integrals in the temperature decomposition approach is to calculate backward trajectories like in Röthlisberger and Papritz (2023). Here, we choose a different method, which does not rely on trajectory calculation but, instead, uses Eulerian tracers.

Our method, which we presented in Mayer and Wirth (2023), is based on the offline advection of passive tracer fields and includes a relaxation term. As a result, one obtains accumulated Lagrangian information at each point of an Eulerian grid at each time step, namely

$$\delta(\boldsymbol{x}, t) = \int_{-\infty}^{t} S(t') e^{-\lambda(t-t')} dt' \ , \qquad (8)$$

where $S(t')$ is some source term and $\lambda$ is the relaxation constant. Note that the term on the right hand side represents a Lagragian integral and features the same exponential weighting as in our decomposition (6). That is, indeed, the primary reason why we introduced the exponential weighting earlier in our $\theta'$ decomposition approach.



To obtain values for $\delta(\boldsymbol{x}, t)$, the tracer method numerically solves the partial differential equation

$$\frac{\partial \delta}{\partial t} = -\boldsymbol{u} \cdot \nabla \delta - \lambda \delta + S \tag{9}$$

with initial condition

$\delta(\boldsymbol{x}, t_{init}) = 0 \,. \tag{10}$

In this equation, $\delta$ is a three-dimensional (tracer) field, which gets advected by the three-dimensional wind $\boldsymbol{u}$ and which is, in addition, subject to some source term $S$. The exponential weighting present in (8) stems from the linear relaxation term $-\lambda \delta$ included on the right hand side of (9). The primary advantage of the relaxation is that it gradually diminishes the influence of past data "on the fly", enabling us to obtain time-continuous Lagrangian information much more efficiently than without

relaxation.

When $S$ can be expressed as the material rate of change of some quantity $a$, i.e., $S = Da/Dt$, (9) can be reformulated into

$$\frac{\partial \psi}{\partial t} = -\boldsymbol{u} \cdot \nabla \psi - \lambda(\psi - a) \quad , \quad \delta(\boldsymbol{x}, t_0) = a \tag{11}$$

in combination with

$$\delta = \psi - a. \tag{12}$$

Instead of $\delta$ itself, the field $\psi$ now constitutes the tracer field and the source term $S$ is impliciltly included in the relaxation term.

A separate tracer is required for each term in the $\theta'$ decomposition. For the horizontal, the vertical, and the seasonal we use formulation (9), while for the diabatic we use formulation (11) in combination with (12). The latter has the significant advantage that we do not require any information about the parcel-based diabatic heating rate, which is often not or not fully

available. For details about the tracer method and its implementation, the interested reader is referred to Mayer and Wirth (2023).

### 2.1.2    Data

This study is based on ERA5 reanalysis data (Hersbach et al., 2020) for the period 2010 to 2022. We use global data on model levels (Hersbach et al., 2017) with a horizontal resolution of $1°$ and a temporal resolution of 3 hours. In the vertical,

we take every second model level between model levels 136 and 50, which cover the entire troposphere and parts of the lower stratosphere. Whenever we show or refer to data on pressure levels, we provide data which have been linearly interpolated from model levels to a set of pressure levels. For the relaxation constant we use $\lambda^{-1} = 7$ days throughout this study.

The variables used are the zonal wind $u$, the meridional wind $v$, the vertical velocity $\dot{\eta}$ that corresponds to the vertical coordinate $\eta$, the pressure-coordinate vertical velocity $\omega$, and pressure $p$. Potential temperature $\theta$ is computed from $p$ and

temperature $T$ as

$$\theta = \left(\frac{p_0}{p}\right)^\kappa T \tag{13}$$





with $\kappa = R/c_p = 0.285$ and $p_0 = 1013.25$ hPa. The climatological average $\bar{\theta}$ in (6) depends both on the time of the year and the time of the day. It is obtained, for each day of the year, by computing a time-of-day specific temporal average over the 13 years considered followed by a 31-day smoothing; effectively, each climatological value is an average over 13 x 31 = 403

individual values.

Output from our method is produced for every 3 hours and subsequently aggregated to yield daily means. Climatological means of the daily mean fields are computed as temporal averages specific for each day of the year, using again a 13-year average followed by a 31-day smoothing. Whenever we use the term "near-surface", we refer to the average over the lowest 50 hPa above the surface.

## 3   Results

### 3.1   A first look at two recent heat waves

In the following we are going to present the results of our decomposition for two recent heat events, namely the Pacific Northwest heat wave 2021 and the UK heat wave 2022. In particular, we discuss the evolution of the near-surface $\theta'$ as well as the time-mean vertical $\theta'$ structure. In doing so, we follow the analysis of Hotz et al. (2024), who applied the temperature

decomposition of Röthlisberger and Papritz (2023) to the aforementioned heat waves. In particular, we use the same definition of the heat wave regions and episodes to enable a streightforward comparison between their results and ours.

#### 3.1.1   The Pacific Northwest heat wave

In 2021, a severe heat wave struck the Pacific Northwest, marking unprecedented temperatures in Canada and the USA. Many cities experienced all-time maximum temperature records broken by several degrees (e.g. Philip et al., 2022; White et al.,

2023). The heat wave was associated with a strong upper-tropospheric quasi-stationary anticyclone (e.g. Philip et al., 2022; Neal et al., 2022), fueled by warm conveyor belt outflow from an upstream cyclone (e.g. Schumacher et al., 2022; Neal et al., 2022; White et al., 2023; Oertel et al., 2023; Röthlisberger and Papritz, 2023; Papritz and Röthlisberger, 2023; Hotz et al., 2024). For this event, we will now showcase the results of our decomposition.

To start with, Fig. 1 illustrates the evolution of the heat wave by showing $\theta'$ and its decomposition for three individual days

during the heat wave. As already noted by previous authors (e.g. Neal et al., 2022; Hotz et al., 2024), the large temperature anomalies first appeared in the upper troposphere and only later emerged near the surface. Initially, a tongue of warm air intruded northward at upper levels (Fig. 1a; cf. Neal et al. 2022), and this can be associated with significant positive contributions from horizontal transport and diabatic heating (Fig. 1b and d) as well as negative contribution from vertical transport (Fig. 1c). Such contribuitions would be expected as resulting from warm conveyor belt outflow. Later, the air mass associated with the

tongue started to form a spiral and became enclosed in the anticyclone (Fig. 1h and i). During the course of the event, the contribution from the pre-existing anomaly seemed to grow steadily (Fig. 1e, j, and o), which first occurred mostly in the upper troposphere and later also near the surface. At the same time, the contributions from horizontal transport, vertical transport,



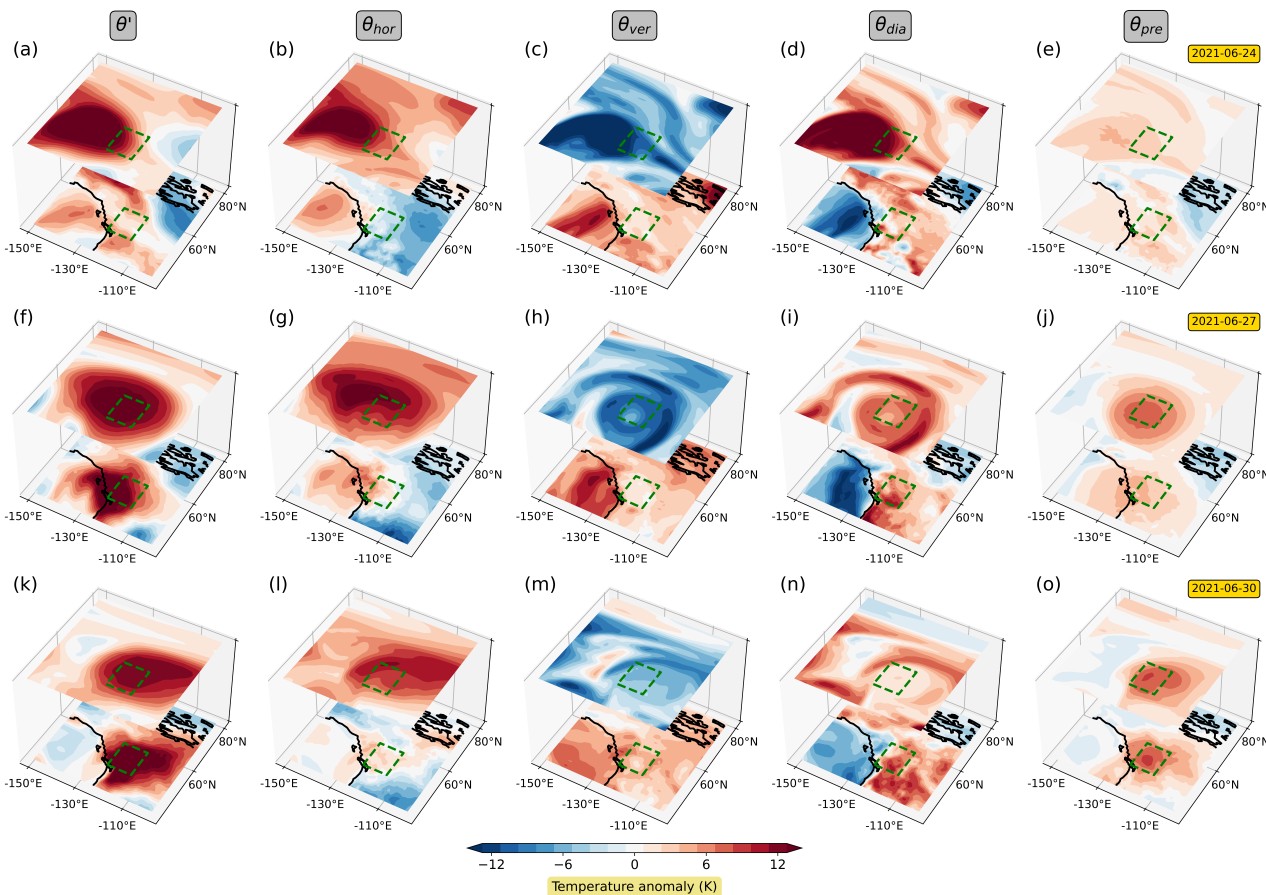

**Figure 1.** Illustration of the 3D structure of $\theta'$ and its contributions from horizontal transport, vertical transport, diabatic heating, and pre-existing anomaly for three selected dates (24 July, 27 July, and 30 July 2021). The upper layer depicts fields averaged between 300 hPa–400 hPa and the lower layer depicts fields near the surface. The green rectangle indicates the Pacific Northwest heat wave region.

and diabatic heating decreased in the upper troposphere. This behavior suggests an effective transfer of heat from the three process terms to the pre-existing term: what initially was classified as horizontal transport, vertical transport, or diabatic heating gradually turned into the pre-existing term. This sequence likely indicates that the large temperature anomalies present at the later stage were generated at least partly before or in the early stage of the heat wave. This interpretation is consistent with the findings by, e.g., Papritz and Röthlisberger (2023), who identify warm-conveyor belt air streams associated with two upstream cyclones as the main heat source for this event.

We next investigate the evolution of the near-surface $\theta'$ over the course of the heat wave. To this end, we show timeseries of the individual $\theta'$ terms near the surface, averaged over the core heat wave region (green box in Fig. 1). Fig. 2 reveals that during the heat event all four decomposition terms contributed positively to $\theta'$, albeit with varying magnitude. Throughout most of the event, the largest positive contribution from the three process terms was given by diabatic heating. The contribution from





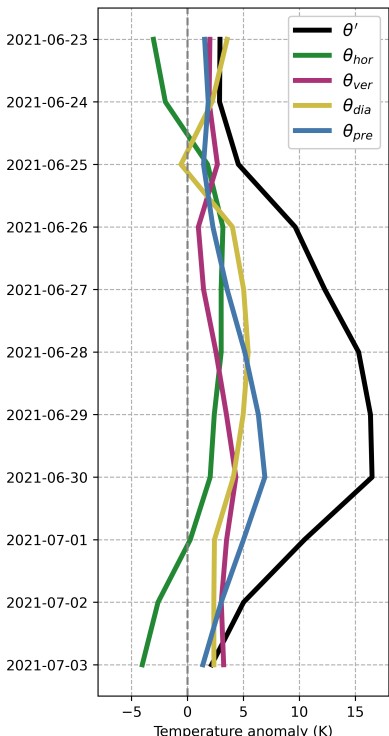

**Figure 2.** Evolution of the near-surface potential temperature anomaly $\theta'$ and its decomposition into contributions from horizontal transport, vertical transport, diabatic heating, and pre-existing anomaly during the Pacific Northwest heat wave. The time series have been obtained by averaging the different terms over the lowest 50 hPa above the surface within the Pacific Northwest heat wave region.

horizontal transport reached its maximum during the early phase and decreased thoughout the course of the event, while the contribution from vertical transport was initially small and gradually increased as the heat wave progressed. Similarly, the pre-
existing anomaly was initially small and grew steadily over the course of the event, until it finally made the largest contribution of all terms during the late stage of the heatwave. Overall, the evolution is similar to that shown by Hotz et al. (2024) (their Fig. 3i and Fig. 4). For example, there is agreement that the near-surface diabatic heating makes the largest contribution of the three process terms throughout the course of the heat wave, even if it is somewhat larger in Hotz et al. (2024) than here. Likewise, in both analyses the contributions from horizontal and vertical transport swap rankings on June 28 in terms of their
positive contribution to $\theta'$.

     Finally, we present the time-mean vertical $\theta'$ structure during the heat wave in Fig. 3. Essentially, these plots show the same characteristics as discussed before with the benefit of full vertical resolution, but in a time-aggregated manner. The general structure of the time-mean vertical fields is, again, similar to the results in Hotz et al. (2024), even though there are differences in the exact values and some structural details. To be sure, we do not expect perfect agreement basically for two reasons: (1)
we use potential temperature rather than temperature as our key variable, and (2) our decomposition includes the pre-existing





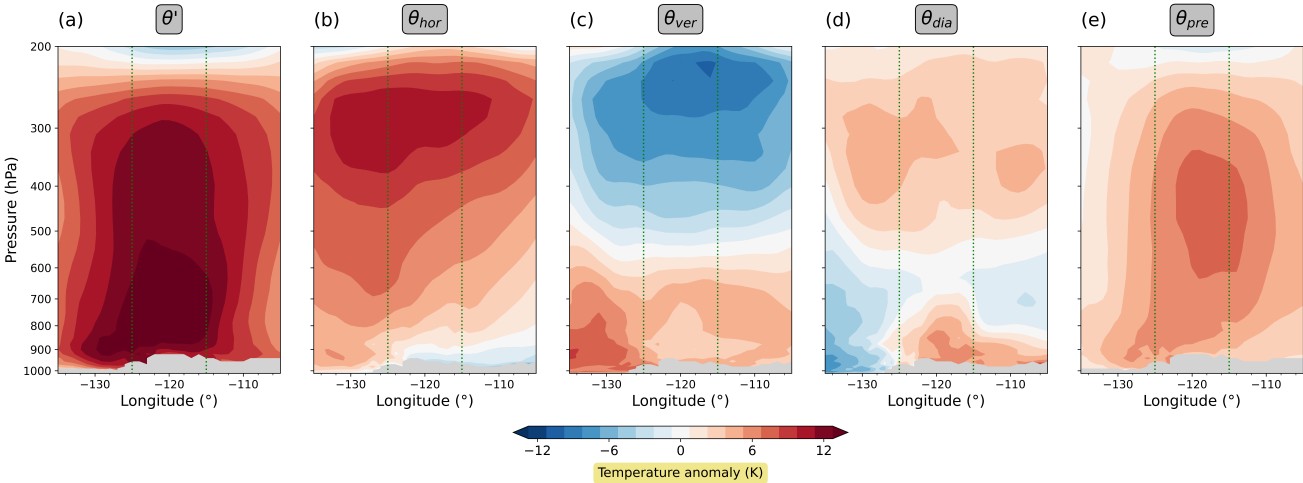

**Figure 3.** Five-day mean (27 June–1 July 2021) vertical cross-section showing the $\theta'$ decomposition during the Pacific Northwest heat wave. Fields are latitudinally averaged between 49–59°N. The panels show (a) the potential temperature anomaly $\theta'$, (b) the contribution from horizontal transport, (c) the contribution from vertical transport, (d) the contribution from diabatic heating, and (e) the contribution from the pre-existing anomaly. The topography is shown in grey. The two green vertical lines in each panel indicate the longitudinal extent of the Pacific Northwest heat wave region.

anomaly which does not exist in the formulation of Röthlisberger and Papritz (2023) due to their specific choice of $t_0$. Item (1) can be expected to be an issue mostly in the upper troposphere where $T$ and $\theta$ deviate substantially; by contrast, item (2) can generally be an issue throughout the atmosphere. During the considered episode, our term $R$ in Fig. 3e maximizes in the middle troposphere. Note that this behavior largely resembles the structure of the Lagrangian age shown by Hotz et al. (2024)
(their Fig. 4b), consistent with our interpretation of $R$ as "pre-existing anomaly".

Overall, our results for the Pacific Northwest heatwave are in good agreement with those of Hotz et al. (2024), suggesting that both methods are able to quantitatively capture the key processes of temperature anomaly formation and their relative importance.

### 3.1.2 The UK heat wave

We now turn to analysing the UK heat wave in 2022. In comparison to the temperature anomalies reached during the Pacific Northwest heat wave, the UK heat wave was less severe. Nonetheless, it was sufficiently strong to break several all-time maximum temperature records in multiple cities across the UK.

Again, we first illustrate the evolution of the heat wave by showing $\theta'$ and its decomposition for three individual days (Fig. 4). Like the Pacific Northwest heat wave, the UK heat wave was also associated with anticyclonic flow, albeit with a much more
transient character. High temperature anomalies first occured in the southwestern parts of Europe, then intensified and crossed the UK, before shifting downstream (Fig. 4a, f, k). As with the Pacific Northwest heat wave, a tongue of warm air initially



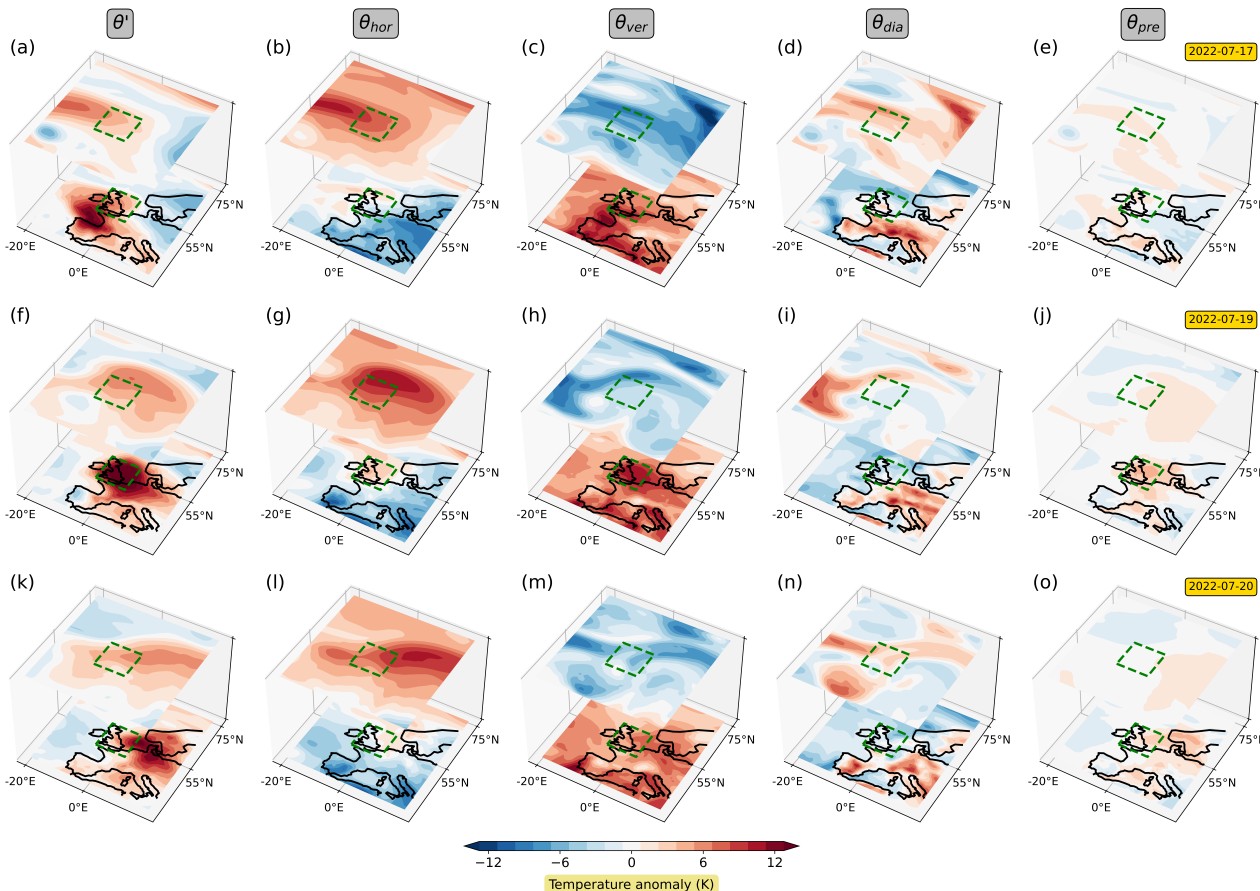

**Figure 4.** Illustration of the 3D structure of $\theta'$ and its contributions from horizontal transport, vertical transport, diabatic heating, and pre-existing anomaly for three selected dates (17 July, 19 July, and 20 July 2022). The upper layer depicts fields averaged between $300 - 400$ hPa, and the lower layer depicts fields near the surface. The green rectangle indicates the UK heat wave region.

appeared in the upper troposphere, accompanied by significant positive contribution from horizontal transport (Fig. 4b), and some positive contribution from diabatic heating (Fig. 4d) as well as negative contribution from vertical transport (Fig. 4c). Presumably, this is again a signature of warm conveyor belt outflow, although the features are overall less coherent than in the Pacific Northwest case. Later, the tongue stretched around the core heat wave region, but unlike the Pacific Northwest heat wave it did not develop a spiralling pattern.

The evolution of the near-surface $\theta'$ decomposition (Fig. 5) reveals that the positive contribution from vertical transport was the highest throughout the event, jointly followed by the contributions from horizontal transport and the pre-existing anomaly. The contribution from diabatic heating, on the other hand, was negative. This is a noticeable difference to the Pacific Northwest heat wave. Our results align quite well with those found by Hotz et al. (2024): they also attribute the largest positive contribution



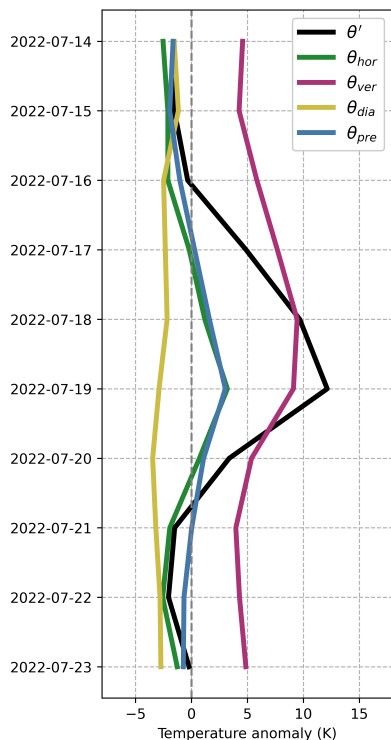

**Figure 5.** Evolution of the potential temperature anomaly $\theta'$ and its decomposition into contributions from horizontal transport, vertical transport, diabatic heating, and pre-existing anomaly during the UK heat wave. The time series have been obtained by averaging the different terms over the lowest 50 hPa above the surface within the UK heat wave region (6°W–4°E, 49–59°N).

to vertical transport, and in both cases the diabatic heating did not make any substantial positive contribution. One noticeable difference is the contribution from vertical transport before and after the heat wave, which in our analysis is considerably larger.

We finally look at the time-mean vertical structure of the event's $\theta'$ decomposition (Fig. 6). Once again, we find small quantitative differences between our analysis and the analysis of Hotz et al. (2024), while the overall patterns of the fields are similar in both analyses. Again, for the reasons mentioned earlier we do not expect perfect agreement between these two analyses. All in all, both analyses lead to essentially the same conclusions, and this suggests that the exact design of the temperature anomaly decomposition and its implementation are of minor importance.

## 3.2 Long-term averages of the terms in the decomposition

We now present long-term averages of the $\theta'$ decomposition for the two regions discussed above, because this will allow us subsequently to adopt a novel perspective on the relevance of the different terms in the decomposition. We believe that this is an important step, because, as we will see shortly, many of the features of the individual terms described above are part of the climatological behavior and, therefore, may not be helpful in "explaining" an anomalous temperature.





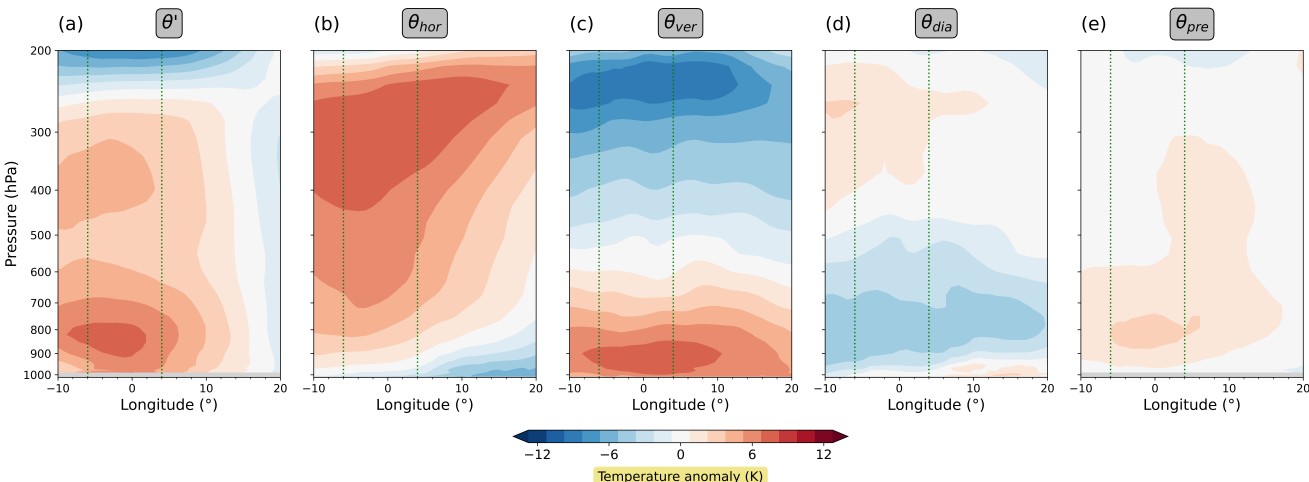

**Figure 6.** Five-day mean (16 July–20 July 2022) vertical cross-sections showing the $\theta'$ decomposition during the UK heat wave. Fields are latitudinally averaged between 49–59°N. The panels show (a) the potential temperature anomaly $\theta'$, (b) the contribution from horizontal transport, (c) the contribution from vertical transport, (d) the contribution from diabatic heating, and (e) the contribution from the pre-existing anomaly. The two green vertical lines in each panel indicate the longitudinal extent of the UK heat wave region.

We start with the long-term averages in the Pacific Northwest region (Fig. 7). Here, the contribution from horizontal transport (Fig. 7a and e) is positive in the upper troposphere and mostly negative close to the surface. This behavior aligns well with the characteristics of the zonal mean Lagrangian circulation (e.g. Townsend and Johnson, 1985; Iwasaki, 1989; Juckes, 2001), which exhibits poleward motion in the upper troposphere (implying a positive contribution from horizontal transport) and equatorward motion near the surface (implying a negative contribution from horizontal transport). Deviations from this overall behavior are most likely due to local land-sea contrasts in temperature. The long-term average of the contribution from vertical transport (Fig. 7b and f) exhibits behavior opposite to that of the horizontal transport, namely a negative contribution in the upper troposphere and a positive contribution near the surface. From a statistical point of view, this makes perfect sense, as air masses close to the surface can only originate from higher above (implying a positive contribution from vertical transport); just below the tropopause the situation is more or less opposite to the extent that the tropopause is a barrier to vertical transport. The contribution from diabatic heating (Fig. 7c and g) is positive in the upper troposphere, possibly due to latent heat release within frequently occurring warm conveyor belts within the North Pacific storm track region. Near the surface, the contribution from diabatic heating is positive over the land and negative over the ocean, which is most likely related to the influence of surface heat fluxes. The contribution from the pre-existing anomaly (Fig. 7d and h) is small throughout the whole troposphere. All terms show little variation over the summer season (Fig. 7i), except for the diabatic one. The latter exhibits a range of about 5 K, with positive values until the mid August and negative values thereafter. Arguably, this behavior reflects the seasonal cycle in solar radiation.



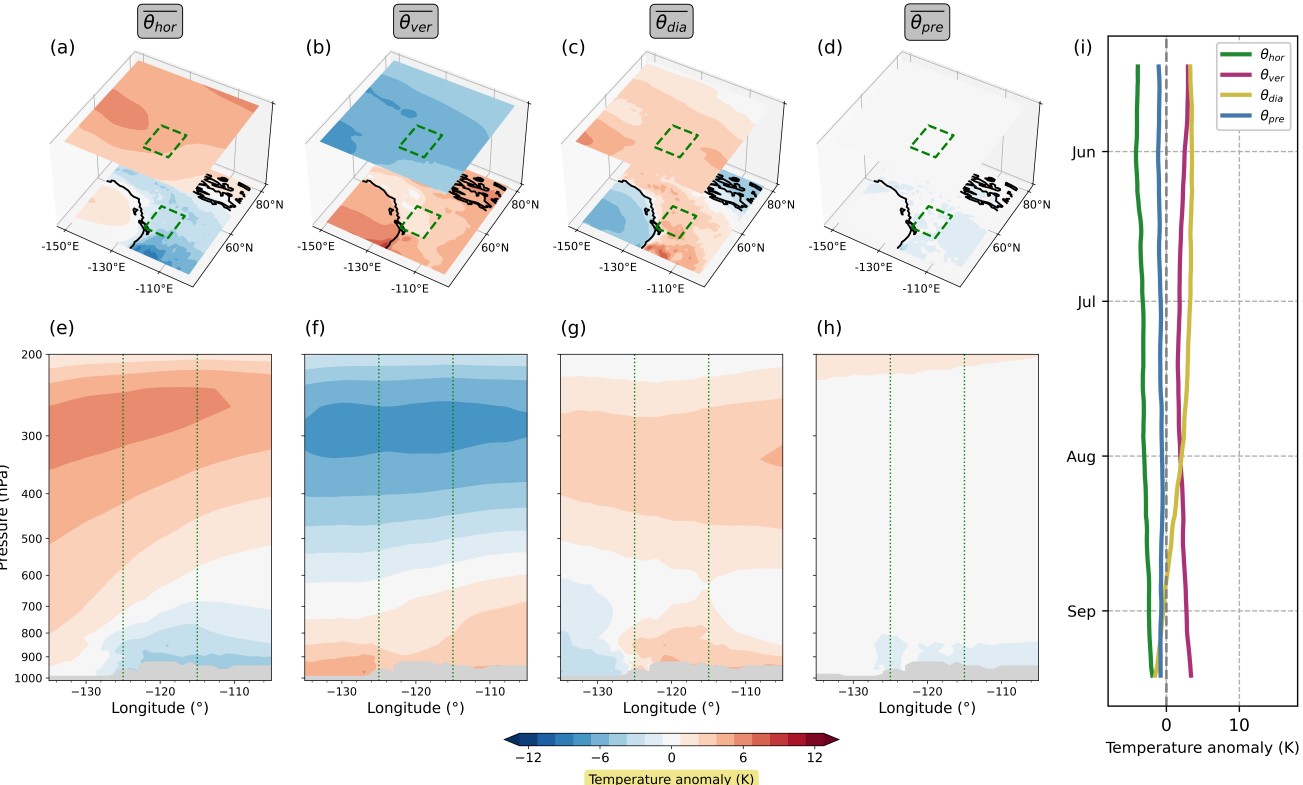

**Figure 7.** Long-term average (2010-2022) of the $\theta'$ decomposition for the Pacific Northwest. Panels (a) to (h) depict the long-term averages during that time of the year the Pacific Northwest heat wave happened (27 June to 1 July). Panel (i) shows the evolution of the near-surface $\theta'$ decomposition during the summer season, averaged over the Pacific Northwest heat wave region. The 3D plots in panels (a) to (d) are analogous to those in Fig. 2b–p) and the vertical cross-sections in panels (e) to (h) are analogous to those in Fig. 3b–e.

The long-term averages of the UK region (Fig. 8) exhibit mostly the same characteristics as those in the Pacific Northwest. The only notable difference manifests near the surface. Here, the contribution from diabatic heating is significantly lower in the UK region compared to the Pacific Northwest, ranging from near-zero to even slightly negative values. Presumably, this feature can be attributed to the region's proximity to the ocean, which implies that air masses located over the UK have a larger chance to originate over the ocean and, therefore, are less exposed to sensible heat fluxes over land.

## 3.3 A second look at the two heat waves from an anomaly-based perspective

We now continue to analyse whether and to what extent the terms in the decomposition were anomalous with respect to their corresponding climatologies. To this end, we compute anomaly fields of the individual decomposition terms and use these in our alternative decomposition. More specifically, we define the anomaly field of a variable $\psi$ as deviation $\psi'$ from its long-term





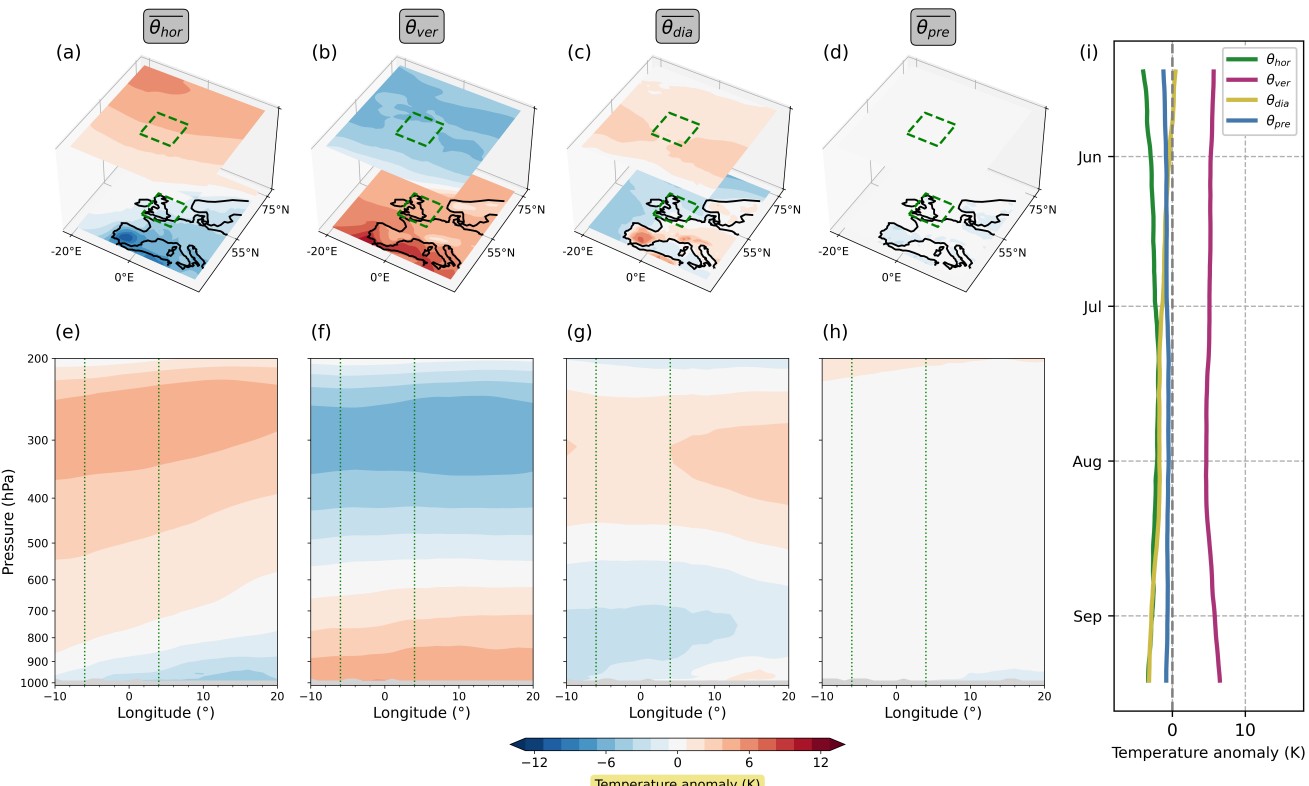

**Figure 8.** Long-term average (2010-2022) of the $\theta'$ decomposition for the UK region. Panels (a) to (h) depict the long-term averages during that time of the year the UK heat wave happened (16 to 20 July). Panel (i) shows the evolution of the near-surface $\theta'$ decomposition during the summer season, averaged over the UK heat wave region. The 3D plots in panels (a) to (d) are analogous to those in Fig. 5b–p) and the vertical cross-sections in panels (e) to (h) are analogous to those in Fig. 6b–e.

average $\overline{\psi}$, i.e.,

$\quad \psi' = \psi - \overline{\psi}\,.$ (14)

Using this definition, the original $\theta'$ decomposition (7) can be expressed as

$$\theta' = (\overline{\theta_{hor}} + \theta'_{hor}) + (\overline{\theta_{ver}} + \theta'_{ver}) + (\overline{\theta_{dia}} + \theta'_{dia}) + (\overline{\theta_{sea}} + \theta'_{sea}) + (\overline{\theta_{pre}} + \theta'_{pre})\,,$$ (15)

where each term is now represented as the sum of its climatological mean and a deviation from this mean. Taking the time-average on both sides, turns (15) into

$\quad \overline{\theta'} = \overline{(\overline{\theta_{hor}} + \theta'_{hor})} + \overline{(\overline{\theta_{ver}} + \theta'_{ver})} + \overline{(\overline{\theta_{dia}} + \theta'_{dia})} + \overline{(\overline{\theta_{sea}} + \theta'_{sea})} + \overline{(\overline{\theta_{pre}} + \theta'_{pre})}\,.$ (16)

Given that the time-averaging scheme is linear, this equation can be rewritten as

$$\overline{\theta'} = (\overline{\overline{\theta_{hor}}} + \overline{\theta'_{hor}}) + (\overline{\overline{\theta_{ver}}} + \overline{\theta'_{ver}}) + (\overline{\overline{\theta_{dia}}} + \overline{\theta'_{dia}}) + (\overline{\overline{\theta_{sea}}} + \overline{\theta'_{sea}}) + (\overline{\overline{\theta_{pre}}} + \overline{\theta'_{pre}})\,.$$ (17)





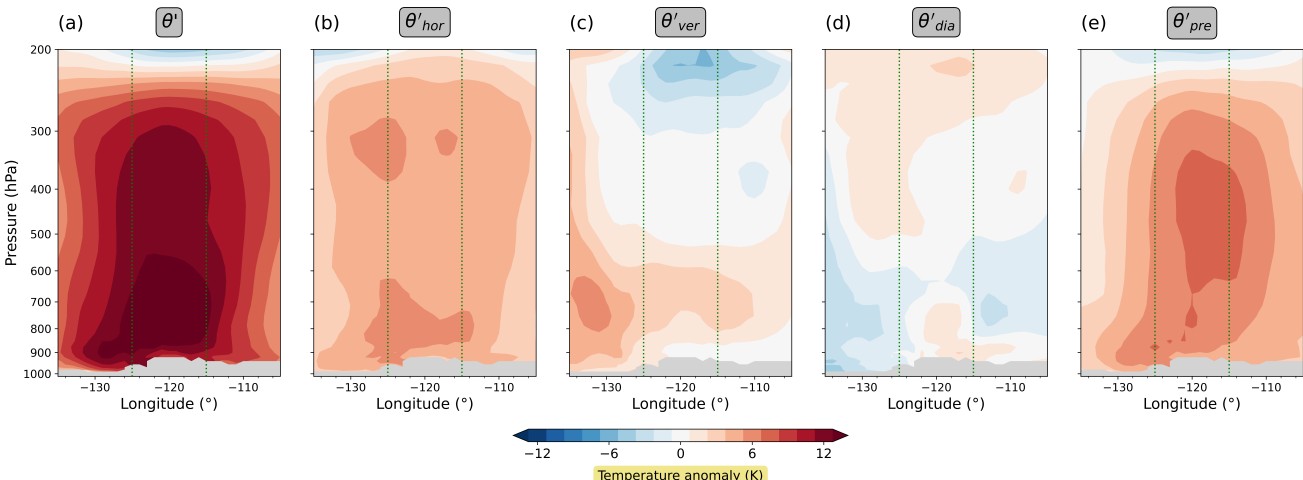

**Figure 9.** Same as Fig. 3, but the $\theta'$ decomposition terms are now depicted as deviations from their long-term averages.

Assuming further that the time-averaging scheme obeys the rules of Reynolds averaging, i.e.,

$$\overline{\psi'} = 0 \quad ; \quad \overline{\overline{\psi}} = \overline{\psi} \tag{18}$$

equation (17) can be simplified to

$$0 = \overline{\theta'_{hor}} + \overline{\theta'_{ver}} + \overline{\theta'_{dia}} + \overline{\theta'_{sea}} + \overline{\theta'_{pre}} . \tag{19}$$

Substituting (19) into (15), yields the following new $\theta'$ decomposition:

$$\theta' = \theta'_{hor} + \theta'_{ver} + \theta'_{dia} + \theta'_{sea} + \theta'_{pre} . \tag{20}$$

In contrast to the original $\theta'$ decomposition (7), which relies on full (absolute) fields, the new $\theta'$ decomposition (20) is based
on anomaly fields. Both formulations are mathematically sound; they differ in that they offer two distinct perspectives on the same subject matter. Note that our daily climatologies imply some 31-day smoothing, such that the assumption of a Reynolds average is strictly speaking not satisfied. However, as we will see below, the corresponding error is negligible.

In the following, we will show that the anomaly-based perspective on the $\theta'$ decomposition leads to substantial differences regarding the relative importance of horizontal transport, vertical transport, and diabatic heating compared to the perspective
in terms of full fields.

### 3.3.1 The Pacific Northwest heat wave revisited

Once again, we begin by discussing the Pacific Northwest heat wave and first look at time-mean vertical cross-sections from the anomaly-based perspective (Fig 9). In the upper troposphere, the contributions from horizontal transport, vertical transport, and diabatic heating are substantially smaller in magnitude than their contributions in terms of full fields. As a consequence, only



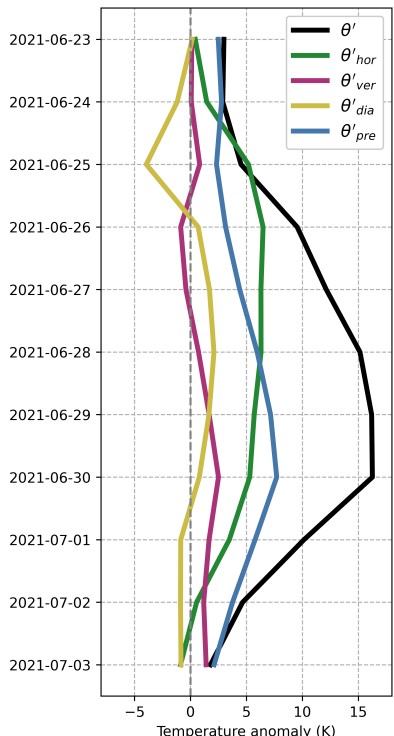

**Figure 10.** Same as Fig. 2, but the $\theta'$ decomposition terms are now depicted as deviations from their long-term averages.

the horizontal transport together with the pre-existing anomaly, still yield a substantial positive contribution to $\theta'$. In the lower troposphere, positive contributions from vertical transport and diabatic heating persist, albeit smaller in magnitude. In contrast, the contribution from horizontal transport changes in such a way that it now provides positive contributions throughout the entire lower troposphere, constituting the largest contribution to $\theta'$ alongside the pre-existing anomaly.

Second, we show the evoultion of the near-surface $\theta'$ decomposition from the anomaly-based perspective (Fig. 10). Like before, all terms yield mostly positive contributions, but the magnitudes differ. Most notably, the diabatic heating no longer dominates among the three process terms; instead, its contribution is significantly smaller than that of horizontal transport throughout the episode. As before, the contribution from vertical transport increases throughout the evolution of the heat wave, but no longer overtakes the contribution from horizontal transport at any point in time. Only the pre-existing term manages to overtake the contribution from horizontal transport at some point, albeit two days later than before.

Thus, the change from the original to the anomaly-based perspective broadly speaking swaps the roles of the diabatic heating and the horizontal transport regarding their contributions to the temperature anomaly $\theta'$. The new perspective takes into account that "normally" the contribution from horizontal transport is negative such that a slight positive contribution in the original decomposition actually represents a rather strong positive contribution in the anomaly-based decomposition. By



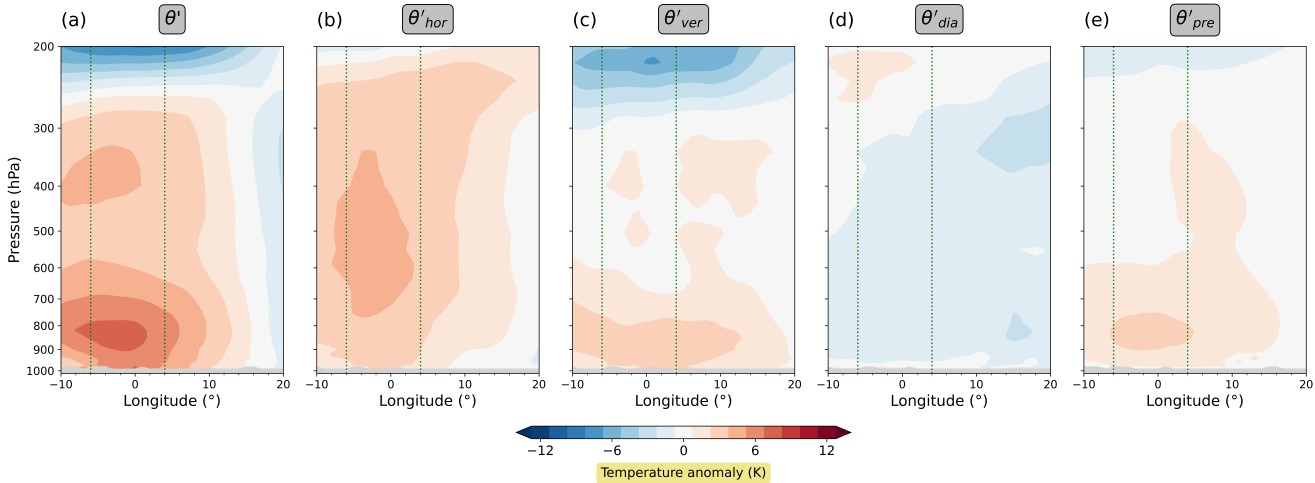

**Figure 11.** Same as Fig. 6, but the $\theta'$ decomposition terms are now depicted as deviations from their long-term averages.

contrast, the diabatic heating, which makes a strong contribution in the original decomposition, turns out to be less important
in the new decomposition because it is just a little stronger than usually.

### 3.3.2 The UK heat wave revisited

Finally, we show the anomaly-based perspective on the UK heat wave. The new decomposition for the time-mean vertical struc-
ture (Fig. 11) also shows substantial differences with respect to the original decomposition. In particular, now the contribution
from vertical transport in the lower troposphere does not dominate the decomposition any longer; instead, the contributions
from horizontal transport and the pre-existing anomaly are of similar magnitude. The same behavior is revealed in the near-
surface time series in Fig. 12.

### 3.4 Statistics for hot extremes in the two heat wave regions

In a final step, we now want to generalize the above results by looking at statistics of heat extremes in the two regions consid-
ered. In this section we will focus on the anomaly-based decomposition.
We choose the following simple definition for heat extreme. We calculate the spatial average of the near-surface potential
temperature anomaly in the respective region for each day between May 15 and September 15 and subsequently select those
days on which the 90th percentile is exceeded. This procedure results in 162 days, which we refer to as "hot days" of the
respective region. Our statistical analysis is then based on these selected hot days.
To start with, Fig. 13 shows vertical cross-sections of $\theta'$ and its (anomaly-based) decomposition averaged over all hot days
in the respective region. In both regions, near-surface hot extremes are most notably associated with an anomalous positive
contribution from horizontal transport throughout the whole troposphere (Fig. 13b and g), contributing around +3 K on average.



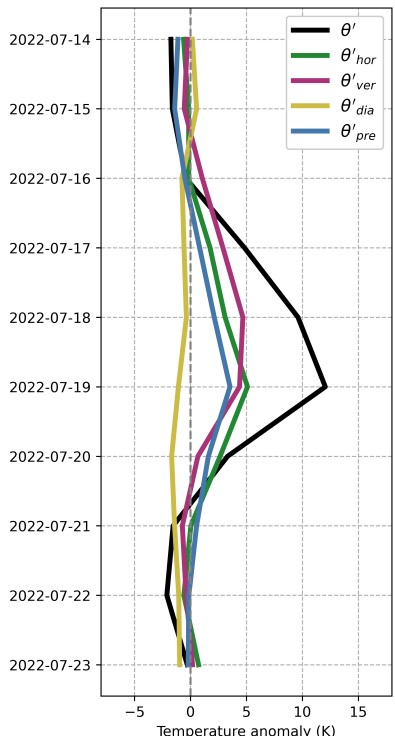

**Figure 12.** Same as Fig. 5, but the $\theta'$ decomposition terms are now depicted as deviations from their long-term averages.

Moreover, as indicated by the contour lines, at least 80% of these extremes show an anomalous positive contribution from horizontal transport up to the tropopause. Additionally, in the lower half of the troposphere the contribution from vertical transport (Fig. 13c and h) is anomalously positive by about 1–2 K on average, reflecting downwelling in anticyclonic flow

during heat waves. The contribution from diabatic heating (Fig. 13d and i) overall tends to be anomalously negative (around -1 K), presumably due to anomalous radiative cooling in strongly subsiding air masses. Pre-existing anomalies (Fig. 13e and j) of 1–2 K on average seem to be nearly always present during hot extremes.

Next, we focus on the near-surface and examine the distributions of the near-surface $\theta'$ decomposition across all hot days in the respective regions (Fig. 14, blue histograms). Most striking is that the distribution of the contribution from horizontal

transport is (nearly) exclusively in the positive range of values in both regions (Fig. 14b and g). This means that (almost) every hot day in the two regions is associated with an anomalous positive contribution from horizontal transport. Similarly, the distribution of the vertical transport, too, shows a clear shift towards positive values (Fig. 14c and h), indicating that hot days are usually (in 94% or 70%, respectively) associated with an anomalous positive contribution from vertical transport. Likewise, the contribution from the pre-existing anomaly is typically positive (Fig. 14e and j).

In contrast, the distribution of the diabatic heating is shifted towards negative values, indicating that hot days tend to be associated with less diabatic heating than average. At first glance this fact may appear paradoxical, given that hot days are





typically characterized by cloud-free skies, intense insolation, and enhanced surface heating. However, one needs to keep in mind that our contributions represent an accumulation over the past few days of the parcel's trajectory, such that the local processes at the parcel's final destination do not necessarily play the dominant role. Instead, it seems that in this case remote effects play a more important role. Most air masses have been unusually warm in their recent past (e.g., Fig. 14e and j) which suggests an anomalously weak transfer of heat from the surface to the atmosphere.

In fact, the above reasoning is consistent with our results from Fig. 14. Apparently, air masses, which have undergone unusually strong diabatic heating (indicated by the green bars), are often linked to anomalously small contributions from horizontal transport; on the other hand, those air masses, which have undergone unusually weak diabatic heating (indicated by orange bars) tend to be associated with anomalously large contributions from horizontal transport. The same qualitative behavior, albeit less pronounced, can be observed for vertical transport and the pre-existing anomaly, respectively.

It is interesting to contrast this general statistical behavior with the situation observed in the Pacific Northwest heatwave. During that episode, clearly positive diabatic heating was observed despite strongly positive contributions from horizontal transport (red bars in Fig. 14b and d). This is consistent with other work suggesting that dry soils contributed to the Pacific Northwest heatwave (e.g. Schumacher et al., 2022; Li et al., 2024). Dry soils can enable unusually strong sensible heat fluxes despite an already warm atmosphere.

Finally, Fig. 15 presents an overview of how frequently each of the $\theta'$ decomposition terms emerges as the dominant one (see figure caption). We provide results for both decompositions to highlight the contrast between the $\theta'$ decomposition in terms of full fields versus anomaly fields. Examining the $\theta'$ decomposition through the full fields (Fig. 15a) reveals that in the Pacific Northwest region vertical transport can be identified as the dominant process in 39% of all hot days, followed by the diabatic heating in 19%. These proportions change drastically when the anomaly-based perspective is adopted (Fig. 15b). Now, it is the horizontal transport – never identified as dominant in the previous perspective – that most frequently emerges as the dominant process. The same behavior, but even more pronounced, can be observed in the UK region (Fig. 15c and d). Note that the results may be sensitive to the precise definition of "dominant", so that the exact numbers may not be interpreted too literally. Nevertheless, the stark qualitative differences apparent in Fig. 15 underscore how strongly the Lagrangian characterization of heat waves depends on the perspective.

## 4   Discussion

One of the key findings of our study is that horizontal transport – if viewed from the anomaly-based perspective – makes a strong, if not the strongest, positive contribution to near-surface hot extremes in the examined regions. What we want to emphasize is that this result does not contradict previous studies which concluded that horizontal advection is negligible (e.g. Bieli et al, 2015; Zschenderlein et al., 2019). Instead, our finding should be viewed as an indication that different perspectives result in different interpretations, although the underlying physics remains the same.

For example, Bieli et al. (2015) analyzed hot extremes over the British isles and found "no substantial meridional transport associated with hot extremes and, in particular, no strong advection of air masses from southerly regions". Naturally, they



**Figure 13.** Vertical cross-section displaying statistical parameters on $\theta'$ and its decomposition (anomaly-based perspective) for hot days in (a)–(e) the Pacific Northwest region and (f)–(j) the UK region. Colors depict the mean, whereas lines depict the fraction of data points that have a positive sign. Fields are latitudinally averaged between 49–59°N. The two green vertical lines in each panel indicate the longitudinal extent of the respective region on which the selection of hot days was based on.



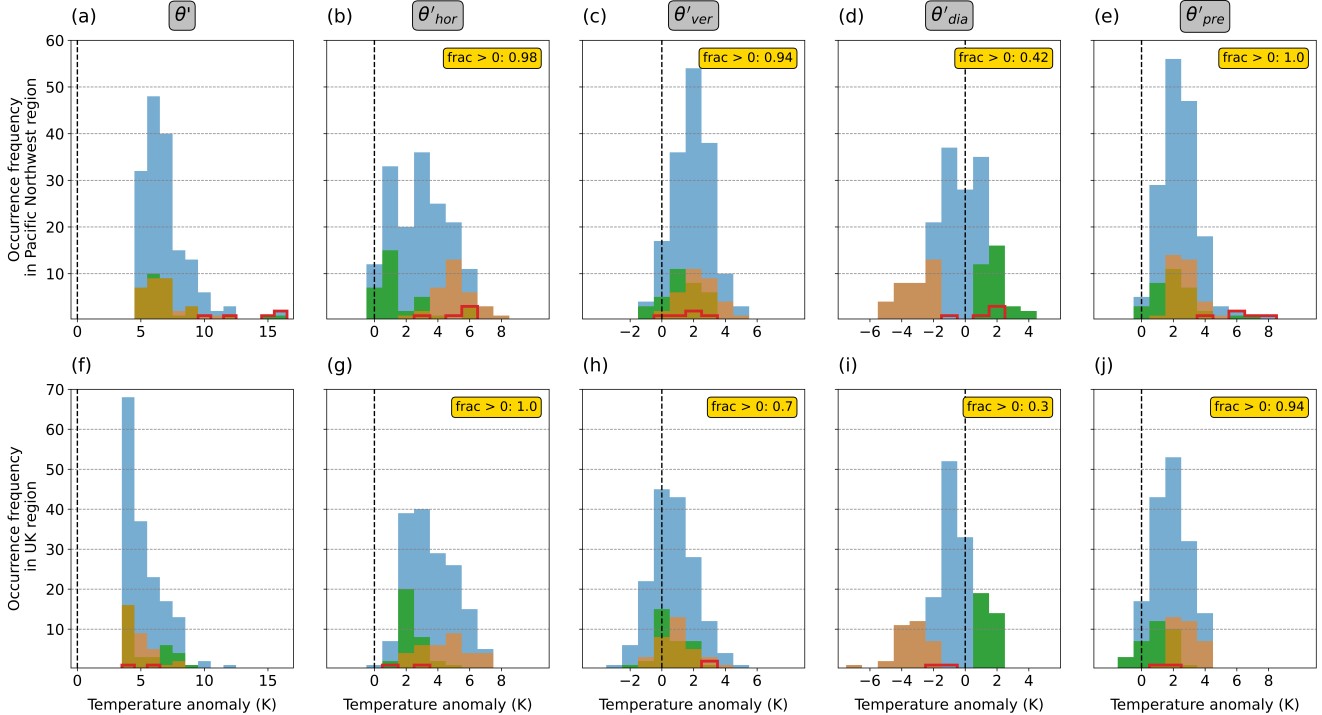

**Figure 14.** Histograms of occurrence frequency of the near-surface $\theta'$ and its (anomaly-based) decomposition for hot days in (a)–(e) the Pacific Northwest region and (f)–(j) the UK region. The histograms are based on area mean values. Blue bars depict all data points. Green bars mark those data points that exceed the 80th percentile in diabatic heating and orange bars mark those data points that are below the 20th percentile in diabatic ehating. Red bars denote data points belonging to the heat wave in 2021 (Pacific Northwest) and in 2022 (UK), respectively. The labels in the top right corner of each panel give the respective fraction of positive values.

concluded from this finding that "horizontal advection of warm air to locations where its temperature represents a large positive deviation from the climatological mean is thus not the primary mechanism in the development of hot extremes". However, based on their observation that air masses during colder days originate significantly further north compared to warm extremes, they could have equally concluded that horizontal advection is relevant for hot extremes in the sense that cold air advection is weaker than usual – and thus anomalously positive. In this context, there seems to be no right or wrong; instead the conclusions

drawn are a matter of interpretation or definition.

To be sure, the idea to compare Lagrangian characteristics of hot air masses with average conditions has been proposed before. For instance, Schielicke and Pfahl (2022) examined variables such as air mass origin, adiabatic heating, and parcel-based diabatic heating for low-level air masses during heat waves and contrasted them with values that would typically occur in summer. After an extensive analysis they found no unusual subsidence nor diabatic heating during heat waves over the

British Isles and concluded that enhanced transport from warm continental regions in the east is of primary importance for the high temperatures. These findings are consistent with our results from the anomaly-based $\theta'$ decomposition, which showed



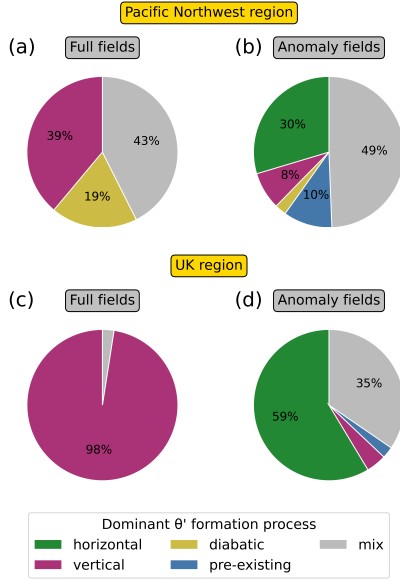

**Figure 15.** Pie charts displaying the relative frequency of the dominant near-surface $\theta'$ formation processes for hot days during summer in (a)–(b) the Pacific Northwest region and (c)–(d) the UK region. A single process is considered dominant if it contributes at least 1.5 times as much as the second-placed term. If no single process is identified as dominant, the day is counted as "mix". The pie charts are based on area mean values. In (a) and (c) the pie charts are based on the $\theta'$ decomposition in terms of full fields, whereas in (b) and (d) the pie charts are based on the $\theta'$ decomposition in terms of anomaly fields.

that horizontal transport from climatologically warmer regions on average contributes more than vertical transport and diabatic heating to a positive near-surface temperature anomaly in the UK region. The strength of our anomaly-based $\theta'$ decomposition is that it provides a systematic and straightforward framework, while other approaches like the one of Schielicke and Pfahl (2022) may be less generic.

In our method, there is one user-defined parameter, namely $\lambda$. In this study, we selected this parameter to be $\lambda^{-1} = 7$ days. We repeated the analysis with $\lambda^{-1} = 3$ days to examine the sensitivity of our results with respect to the value of $\lambda$. In the decomposition in terms of full fields, the contributions from horizontal transport, vertical transport, and diabatic heating were generally smaller in magnitude with $\lambda^{-1} = 3$ days compared to the analysis with $\lambda^{-1} = 7$ days. On the other hand, the contribution from the pre-existing anomaly was generally larger with $\lambda^{-1} = 3$ days compared to the analysis with $\lambda^{-1} = 7$ days.

This behavior can be broadly expected: the analysis with a smaller $\lambda^{-1}$ captures a smaller fraction of the processes that are accounted for by the larger $\lambda^{-1}$, attributing the rest to the pre-existing anomaly. In other words, what used to be contributions from horizontal transport, vertical transport, or diabatic heating is more likely to be transferred into contributions from the pre-existing anomaly. However, and most importantly, in both heat waves the overall vertical structure of the decomposition





terms was similar for both values of $\lambda$. Similarly, in the anomaly-based decomposition only small differences between both values of $\lambda$ occurred. This suggest that, overall, the exact value of $\lambda$ is of secondary importance.

A further characteristic of our temperature diagnostics is that the pre-existing anomaly usually exhibits the same sign as the actual temperature anomaly. This pattern holds true not only for positive temperature anomalies (see heat wave examples and statistical analysis), but also for negative temperature anomalies (not shown). At first glance, this behavior might seem

disappointing, as our method often fails to completely attribute the observed temperature anomaly to the three physical processes. On the other hand, the existence of a pre-existing anomaly may be the reflection of a very fundamental atmospheric property, namely its persistence. Persistence is a well known phenomenon over a wide range of time scales and variables (e.g. Pelletier, 1997; Pelletier and Turcotte, 1997; Koscielny-Bunde et al., 1998). Persistence of temperature, for instance, means that warm days (or months or years) are, more often than not, followed by warm days and cold days by cold days. In other

words, persistence means that a variable (e.g. temperature) exhibits autocorrelation. This autocorrelation is particularly strong on the synoptic time-scale (e.g. Talkner and Weber, 2000; Eichner et al., 2003). Considering the autocorrelation of temperature, it is not surprising that the pre-existing anomaly tends to be (strongly) positive during heat waves: If there is a large temperature anomaly present today, it is very likely that a temperature anomaly had already existed in the vicinity a few days earlier. Therefore, we do not consider our pre-existing term and its behavior as a weakness of our method, but rather as a term

with genuine physical significance, providing in some sense true added value.

Lastly, it is important to acknowledge that Lagrangian methods are generally subject to errors (e.g. Stohl, 1998; Stohl et al., 2002; Mayer and Wirth, 2023). This caveat applies to our analysis just as much as it applies to most previous trajectory-based studies of near surface hot extremes (e.g. Harpaz et al., 2014; Bieli et al., 2015; Zschenderlein et al., 2019; Spensberger et al., 2020; Catalano et al., 2021; Hochman et al., 2021; Schielicke and Pfahl, 2022; Röthlisberger and Papritz, 2023; Hotz et al.,

2024). One prominent issue is the atmospheric boundary layer, where turbulent mixing and convection are particularly pronounced. These processes are not fully accounted for when simply using mean wind fields from reanalysis data. Nevertheless, we believe that Lagrangian analyses such as ours and previous ones do provide valuable information when carefully interpreted. For instance, we never show results for a single grid point, rather we provide information that is aggregated over time or space. The premise would be that this aggregation reflects the mean behavior of real air parcels to a fair approximation.

While this assumption may seem strong, questioning it would also cast doubt on the validity of most other trajectory-based studies. Moreover, presumably most of the air parcels spend at least some time of their recent history in the free atmosphere, where uncertainties of Lagrangian information are likely to be much smaller.

## 5  Summary and Conclusions

In this paper, we analyzed heat waves from a Lagrangian perspective. Core to our Lagrangian analysis was a Lagrangian $\theta'$

decomposition based on the $T'$ decomposition proposed by Röthlisberger and Papritz (2023). Our Lagrangian $\theta'$ decomposition partitions a potential temperature anomaly at a given location into contributions from horizontal transport, vertical transport,



diabatic heating, and a pre-existing temperature anomaly in order to quantify the relative importance of these processes in the formation of heat waves.

To obtain the Lagrangian information required for the analysis, we used the method of tracer advection with relaxation introduced by Mayer and Wirth (2023). This sets us apart from the majority of other Lagrangian studies on heat waves, as we do not rely on trajectory calculations. The tracer method offers the advantage that long periods of data can be processed efficiently, facilitating the computation of climatologies.

First, we applied a decomposition closely resembling the original decomposition by Röthlisberger and Papritz (2023). We presented the results of that decomposition for two recent heat events, namely the Pacific Northwest heat wave in 2021 and the UK heat wave in 2022. In this context, the preceding study of Hotz et al. (2024), who also used the decomposition by Röthlisberger and Papritz (2023), served as a template and reference. In both heat waves, we observed that the upper troposphere experienced positive contributions from horizontal transport and diabatic heating, which were partly compensated by negative contributions from vertical transport. Near the surface, during the Pacific Northwest heat wave diabatic heating dominated most of the event, whereas during the UK heat wave vertical transport was clearly dominant. Overall, these results are consistent with the results of the reference study and differ only in details.

Next, we provided, for the first time, long-term averages of the respective terms in the temperature anomaly decomposition. Interestingly, these averages showed similar characteristics to those observed during the heat wave periods. For instance, we observed that in the Pacific Northwest, air masses near the surface typically undergo significant diabatic heating, while in the UK, they experience a substantial positive contribution from vertical transport. This suggested that the notable positive contribution from diabatic heating (oberserved during the Pacific Northwest heat wave) or from vertical transport (oberserverd during the UK heat wave) may not have been the decisive factor for the unusually high temperatures. Their contributions were substantial, though not exceptionally so. This motivated us to introduce a new temperature anomaly decomposition based on the *anomaly* fields of the respective terms.

Compared to the decomposition in terms of full fields, the anomaly-based decomposition led to significant differences regarding the relative importance of the respective terms. For example, during the Pacific Northwest heat wave diabatic heating no longer constituted one of the largest contributions, but clearly was second to the contribution from horizontal transport. Similarly, in the UK heat wave vertical transport no longer stood out as the sole dominant factor; instead, horizontal transport contributed significantly, too. Loosely speaking, near the surface the horizontal transport thus emerged as the "winner" of this shift in perspective.

Finally, we complemented our study by a statistical analysis of hot days in the respective regions. Most notably, this analysis revealed that hot days in both regions consistently coincide with abnormally positive contributions from horizontal transport near the surface, often extending all the way to the tropopause. This suggests that anomalously positive horizontal transport can be regarded as a necessary (but possibly not sufficient) prerequisite for the formation of a hot extreme.

We believe that a lot can be learned from the anomaly-based decomposition, as it inherently provides an intuitive comparison to what can be expected from climatology. However, we are aware that even the anomaly-based decomposition, which may be regarded as an extension of the decomposition originally suggested by by Röthlisber and Papritz, does not provide answers





to all questions. For instance, while we can learn from the anomaly-based perspective that anomalously positive horizontal transport seems to be an inevitable ingredient for the formation of unusually warm temperatures, it does not fully elucidate why certain warm events are more extreme than others. For instance, dryer-than-usual soils may be a contributing factor. How-
ever, their effect would be clearly reflected in an anomalously strong diabatic heating only if other variables, such as horizontal transport, were not affecting the diabatic heating, too. Conditioning on specific factors, such as prescribed dynamics (i.e., prescribed horizontal and vertical transport), might offer a potential avenue to disentangle such effects. Nonetheless, the existing correlations between the respective variables in the decomposition still pose a challenge for drawing causal conclusions.

In summary, the paper demonstrated that various ways exist to perform meaningful temperature anomaly decompositions. At
the same time, this means that there is no unique answer regarding the extent to which horizontal transport, vertical transport, and diabatic heating contribute to a given temperature anomaly. Ultimately, the answer to this question seems to be a matter of perspective.

*Code and data availability.* All results are based on the ERA5 reanalysis from ECMWF. The Python code for the tracer method, along with the scripts and data used for the analysis, are available from the authors upon request.

**Appendix A: Reformulation of the expression for $\theta'$**

Let $\tau$ be the sum of all terms that bring about a change in a parcel's potential temperature anomaly $\theta'$, i.e.,

$$\tau = \frac{D\theta}{Dt} - \frac{\partial\bar{\bar{\theta}}}{\partial t} - \boldsymbol{v}\cdot\nabla_h\bar{\theta} - \omega\cdot\frac{\partial\bar{\theta}}{\partial p} \,, \tag{A1}$$

such that

$$\frac{D\theta'}{Dt} = \tau \,. \tag{A2}$$

Integrating $\tau$ along the parcel's trajectory yields

$$\theta' = \int_{t_0}^{t} \tau dt' + \theta'_0 \,. \tag{A3}$$

where $t_0$ is the start time of the integration, t is the time of interest, and $\theta'_0$ is the temperature anomaly at start time. The integral in (A3) can be extended to start at $-\infty$ if, at the same time, that part of the integral is subtracted which arises from the time interval $(-\infty, t_0)$:

$$\theta' = \int_{-\infty}^{t} \tau dt' - \int_{-\infty}^{t_0} \tau + \theta'_0 \,. \tag{A4}$$

The latter equation is equivalent to

$$\theta' = \int_{-\infty}^{t} \tau e^{-\lambda(t-t')} dt' + \int_{-\infty}^{t} \tau \left(1 - e^{-\lambda(t-t')}\right) dt' - \int_{-\infty}^{t_0} \tau dt' + \theta'_0 \,, \tag{A5}$$



where we introduced the exponential weighting that is crucial to allow the application of our tracer method. Defining

$$R = \int_{-\infty}^{t} \tau \; \left( 1 - e^{-\lambda(t-t')} \right) dt' \tag{A6}$$

and positing that

$$\int_{-\infty}^{t_0} \tau \, dt' = \theta_0' \tag{A7}$$

one obtains

$$\theta' = \int_{-\infty}^{t} \tau e^{-\lambda(t-t')} dt' + R \,. \tag{A8}$$

*Author contributions.*  Amelie Mayer carried out the analysis with some advice by Volkmar Wirth. Both authors jointly wrote the paper.

*Competing interests.*  None of the authors has any competing interests.

*Acknowledgements.*  The research leading to the results in this article has been performed within the subproject C4 "Predictability of European heat waves" of the Transregional Collaborative Research Center SFB/TRR 165 "Waves to Weather" (https://www.wavestoweather.de) funded by the German Research Foundation (DFG). We thank the Copernicus Climate Change Service for granting free access to the ERA5 data. We made occasional use of ChatGPT (version 3.5) to refine sentence structures and improve formulations.



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
