# Peer review of "Two different perspectives on heat waves within the Lagrangian framework"

_EGUsphere, 2024_

## Referee Comment (RC1)

Review: Lagrangian characterization of heat waves: The perspective matters

This paper quantifies the processes leading to two major heatwaves using two different methods. While both methods use Lagrangian information to decompose the temperature anomaly into different processes, the second method isolates the anomalous contribution of each process from its climatological value to the heat accumulation. Notably, the authors show that when considering the anomalous contributions, horizontal advection plays a larger role in near-surface heat accumulation than the total decomposition suggests. This paper presents a new framework for determining the relative importance of different processes to heatwave formation, and I would be happy to recommend it for publication with major revisions.

**Main Comments:**

I'm unclear as to what the $\theta_{pre}$ term represents physically. Does this capture the memory of the heatwave on daily timescales? Including some discussion of the physical meaning of this term in the methods would be useful.

The explanation of the Eulerian tracers was difficult to follow. I understand the full explanation is provided in a previous publication, but it would be helpful to show how $\theta$ enters into equations (9) and (11) to provide some physical intuition into how the Lagrangian information is obtained. Additionally, including an explanation why the Eulerian method was chosen instead of the direct trajectory calculations in the methods section would be useful.

I would appreciate more clarification on how the long-term averages of the decomposed potential temperature contributions are calculated in section 3.2 (i.e., the data shown in Figs. 7 & 8).

**Minor Comments:**

It would be helpful to explicitly state in the text which dates are considered part of the heatwave for both cases examined.

Line 91: There seems to be a $\bar{\theta}$ missing from the first term in R.

Line 138: Is there a reason only the years 2010 – 2022 were included in the time-averages?

Line 234: Stating the years included in the long-term averages would be helpful here. It is included in the caption of Fig. 7, but should also be stated in the text.

Line 255-260: It is stated that the contribution from diabatic heating in the UK differs from in the Pacific Northwest due to the UK is in closer proximity to the ocean. However, the Pacific Northwest is also close to the ocean, with the mean flow likely originating over the ocean. Is the difference that the UK is surrounded by ocean?

Line 335-336: Does the negative shift in diabatic heating indicate that surface fluxes are smaller than average on hot days, or that the air parcels are not close enough to the surface to be affected by surface fluxes for the majority of their trajectory?

Fig. 13: What is the bold black line on these figures?  It seems to be the 50% contour, but it doesn't extend across the domain in every panel.

Fig. 15: Please make this figure bigger and perhaps bold the text on the pie charts.  The percentages are difficult to read.

There are several misspellings in the text.

---

## Author Comment (AC1)

**Response to Reviewers**

corncerning the manuscript with original title

**Lagrangian characterization of heat waves: The perspective matters**

and revised title

**Two different perspectives on heat waves within the Lagrangian framework**

by A. Mayer and V. Wirth,

*submitted to Weather and Climate Dynamics*

We thank both reviewers for their careful reading and their valuable feedback. We tried to address all the issues raised by the reviewers as detailed in this reply document, where we provide answers (in blue) to the reviewer's comments (in black). Line numbers in our reply refer to those in the revised manuscript.

**Reviewer 1**

This paper quantifies the processes leading to two major heatwaves using two different methods. While both methods use Lagrangian information to decompose the temperature anomaly into different processes, the second method isolates the anomalous contribution of each process from its climatological value to the heat accumulation. Notably, the authors show that when considering the anomalous contributions, horizontal advection plays a larger role in near-surface heat accumulation than the total decomposition suggests. This paper presents a new framework for determining the relative importance of different processes to heatwave formation, and I would be happy to recommend it for publication with major revisions.

**Main Comments:**

I'm unclear as to what the $\theta_{pre}$ term represents physically. Does this capture the memory of the heatwave on daily timescales? Including some discussion of the physical meaning of this term in the methods would be useful.

Thanks for this remark. To further clarify the physical meaning of $\theta_{pre}$, we have now included the following lines (Lines 106-109) in the methods section: "The last term, $\theta_{pre}$ , corresponds to the residuum R and will be referred to as pre-existing potential temperature anomaly in the following. It reflects the accumulation along the parcel's trajectory of the three process terms from earlier times, i.e., up to about $\lambda^{-1}$ before the considered point in time t. In other words, $\theta_{pre}$ is analogous to the

initial potential temperature anomaly $\theta_0'$ of a trajectory of length $\lambda^{-1}$ ." In some sense, $\theta_{pre}$ indeed captures the memory of the heat wave. The term "remembers" the potential temperature anomaly of the air parcel from about $\lambda^{-1}$ earlier. However, it "forgets" the specific details of how this potential temperature anomaly developed, meaning it does not "remember" which of the three process terms contributed to the temperature anomaly and to what extent; it only captures their accumulated effect.

The explanation of the Eulerian tracers was difficult to follow. I understand the full explanation is provided in a previous publication, but it would be helpful to show how $\theta$ enters into equations (9) and (11) to provide some physical intuition into how the Lagrangian information is obtained. Additionally, including an explanation why the Eulerian method was chosen instead of the direct trajectory calculations in the methods section would be useful.

We agree that the explanation of how we compute the terms in the temperature anomaly decomposition using the tracer method has fallen somewhat short. We extended the explanation (Lines 140-156) by explicitly stating the used source terms $S$ in formulation (9) of the tracer method and the term $a$ in formulation (11) and (12). Additionally, we provide specific examples of the partial differential equations we solve, making it clearer how $\theta$ enters into the equations. Further, we included a paragraph (Lines 111-119) discussing the stength of the tracer method and explaining why we decided to use this method instead of trajectory calculations.

I would appreciate more clarification on how the long-term averages of the decomposed potential temperature contributions are calculated in section 3.2 (i.e., the data shown in Figs. 7 & 8).

Initially, we described the computation of the long-term averages of the decomposed potential temperature contributions in Section 2. However, in response to the reviewer's comment, we realized that this may not have been the most appropriate place to provide this information. We have moved the description of the computation of the long-term averages to Section 3.2 (Lines 260-263) now and revised it for clarity. Additionally, we have added a further sentence to the captions of Figures 7 and 8 to clarify what exactly is shown there.

**Minor Comments:**

It would be helpful to explicitly state in the text which dates are considered part of the heatwave for both cases examined.

In Line 182 we now explicitly state which days are considered part of the heat wave.

Line 91: There seems to be a $\bar{\theta}$ missing from the first term in R.

Thanks for spotting. We corrected this error.

Line 138: Is there a reason only the years 2010 – 2022 were included in the time-averages?

Yes, there are two reasons why we (only) included the years 2010-2022 in the time averages.

The first reason is that we wanted to use a similar time period as Hotz et al. in order to compare our results properly to their results. The temperature climatology used in their study is based on a 9-year period, which is comparable to the 13-year period we chose. We now mention this in Lines 171-172.

The second reason is that we wanted to ensure that the anomalies we present can be considered anomalous compared to today's conditions rather than to those from the past. I.e. we deliberately wanted to avoid capturing signals of climate change. In the end the chosen length of the period was a compromise between robustness of statisctis and stationarity of the climate. We now explicitly mention this in the text (Lines 263-267).

Line 234: Stating the years included in the long-term averages would be helpful here. It is included in the caption of Fig. 7, but should also be stated in the text.

We now explicitly state the years in Line 261.

Line 255-260: It is stated that the contribution from diabatic heating in the UK differs from in the Pacific Northwest due to the UK is in closer proximity to the ocean. However, the Pacific Northwest is also close to the ocean, with the mean flow likely originating over the ocean. Is the difference that the UK is surrounded by ocean?

Thanks for this remark. We agree that the formulation "in proximity to the ocean" may be misleading. Therefore, we rephrased the sentence from "Presumably, this feature can be attributed to the region's proximity to the ocean, ..." to "Presumably, this feature can be attributed to the fact that the UK is surrounded by the ocean, ..." (see Lines 287-289).

Line 335-336: Does the negative shift in diabatic heating indicate that surface fluxes are smaller than average on hot days, or that the air parcels are not close enough to the surface to be affected by surface fluxes for the majority of their trajectory?

Thank you for this thoughtprovoking question, which turned out to be more challenging to answer than it may appear at first. In addressing it, we examined the surface sensible heat fluxes during hot days in the respective regions and found they were lower than average in both regions. This suggests that the negative shift in diabatic heating is indeed an indication for smaller-than-usual surface heating. However, we also found anomalous positive temperature anomly contributions from vertical transport. This implies that these air masses originate from higher altitudes than usually, suggesting in turn that they spend less time near the surface where they could be influenced by surface fluxes. Thus, the negative shift in diabatic heating could also indicate that the air parcels are not close enough to the surface to be affected by surface fluxes for the majority of their trajectory. Most probably, the negative shift in diabatic heating indicates both. However, distinguishing between the two effects is quite complex, as they are to some extent intrinsically correlated. A stronger than usual subsidence implies a higher than usual initial potential temperature, resulting in a smaller than usual surface heating due to a smaller than usual temperature gradient between the surface and the atmosphere under otherwise similar surface conditions. Does this mean that the lower diabatic heating is a result of reduced sensible heat fluxes or altered dynamics? Honestly we cannot tell. This is the difficulty we encounter throughout the entire analysis.

In response to this question, we revised Lines 366-376.

Fig. 13: What is the bold black line on these figures? It seems to be the 50% contour, but it doesn't extend across the domain in every panel.

Yes, the bold black line should represent the 50% contour. We had a plotting issue here. The line now extends across the full domain in every panel.

Fig. 15: Please make this figure bigger and perhaps bold the text on the pie charts. The percentages are difficult to read.

In fact, the figure was quite small. We have enlarged the figure and increased the font size of the percentages.

There are several misspellings in the text.

Thanks for mentioning. We corrected all misspellings we found.

**Reviewer 2**

**General Comments**

The article presents a detailed analysis of the mechanisms driving summer heat waves, comparing perspectives in the Lagrangian framework. The study is well-structured and contributes valuable insights into the field. However, several areas could benefit from further clarification, additional references, and some adjustments to the presentation of data. Below are specific comments that should be addressed to improve the manuscript.

**Specific Comments**

- **Line 25 - Citation Addition:**

**Comment:** The study would benefit from adding the reference Garfinkel et al. (2024) when discussing horizontal advection.

**Suggested Action:** Please include the following citation:
Garfinkel, C.I., Rostkier-Edelstein, D., Morin, E., Hochman, A., Schwartz, C. & Nirel, R. (2024). Precursors of summer heat waves in the Eastern Mediterranean. *Quarterly Journal of the Royal Meteorological Society*, 1–17. Available from: https://doi.org/10.1002/qj.4795

Thanks for this suggestion. We included the citation.

- **Line 42-43 - Use of LAGRANTO or HYSPLIT:**

**Comment:** Is there a specific reason for not utilizing LAGRANTO or HYSPLIT in your analysis? These tools are widely used for tracking air parcels.

**Suggested Action:** Briefly discuss why alternatives like LAGRANTO or HYSPLIT were not considered.

In response to this comment we added a full paragraph (Lines 111-119) discussing the strength of the tracer method and explaining why we decided to use this method instead of trajectory calculations.

- **Figures 2 and 5 - Orientation Adjustment:**

**Comment:** The current layout of Figures 2 and 5 may be challenging to interpret. I recommend changing the orientation to have the x-axis represent time.

**Suggested Action:** Modify the figures to have time on the x-axis, which could make the temporal patterns more apparent.

We had time on the y-axis to be consistent with the figure from Hotz et al. (2023). However, we agree that it is more intuitive to put time on the x-axis. Therefore we modified Figures 2 and 5 as suggested by the reviewer.

- **Period of Study - Influence of Global Warming:**

**Comment:** The period of 2010-2022, which coincides with significant global warming, might influence your results. Is there a specific reason for selecting this timeframe?

**Suggested Action:** Address whether the chosen period might affect the study's conclusions, and if possible, justify the selection of this period or discuss its implications.

There are two reasons why we (only) included the years 2010-2022 in the time averages.

The first reason is that we wanted to use a similar time period as Hotz et al. (2023) in order to compare our results properly to their results. The temperature climatology used in their study is based on a 9-year period, which is comparable to the 13-year period we chose. We now mention this in Lines 171-172.

The second reason is that we wanted to ensure that the anomalies we present can be considered anomalous compared to today's conditions rather than to those from the past. I.e. we deliberately wanted to avoid capturing signals of climate change. In the end the chosen length of the period was a compromise between robustness of statisctis and stationarity of the climate. We now mention this in the text (Lines 263-267).

- **Use of the Term Climatology:**

**Comment:** The term "climatology" typically refers to a 30-year period. Using it for a 13-year dataset might not be appropriate.

**Comment:** Were the studied heat waves used to compute the long-term mean? Since the period is rather short, this may have influenced the results.

**Suggested Action:** Consider revising the terminology or discuss whether using a 30-year climatology would alter your findings.

We appreciate the reviewer's concerns regarding our decision to include the studied heat waves in the computation of the long-term mean. While it is true that excluding the heat wave periods would lead to some differences in the results, we believe that these differences would be relatively small and that the core message of our paper would not be affected. Furthermore, we hold the view that extreme events such as heat waves should be considered when calculating climatological averages, as they are an essential aspect of the true climatological behavior.

We now state that the considered period "is admittedly shorter than the traditional definition of a climatological mean" (Line 263-264) and we now discuss why we use this rather short period to compute the climatological average (see our answer from above).

- **Typo in Figure's 14 caption:**

**Comment:** There is a typographical error in Figure 14, where "ehating" should be corrected to "heating."

**Suggested Action:** Please correct this typo.

Thanks for spotting. We corrected this typo.

- **Line 404 - Reference Addition:**

**Comment:** It would be beneficial to add the reference to Hochman et al. (2021) when discussing the persistent definition of heat waves.

**Suggested Action:** Include the following reference:
Hochman et al. (2021).

Thank you for your suggestion. We appreciate the insights presented in this interesting paper and have given the reference careful consideration. However, we ultimately decided not to include it, as we believe it may not fully align with the context of our work. By persistence, we mean that an anomalously warm (or cool) day is typically followed by another anomalously warm (or cool) day. In contrast, Hochman et al. (2021) quantify the degree to which hot or cold periods exhibit more or less persistence compared to other periods. We feel that we are referring to a much simpler concept than what Hochman et al. discuss. Therefore, we believe that including the reference here might be

misleading. Should we be mistaken, we would greatly appreciate the reviewer sharing their thoughts, and we would be happy to reconsider the reference.

- **Discussion on Lagrangian and Eulerian Perspectives:**

**Comment:** Garfinkel et al. (2024) used an Eulerian approach, whereas Röthlisberger and Papritz (2023) used a Lagrangian one, leading to slightly different conclusions about heat wave drivers. Discussing these differing perspectives could enhance your analysis.

**Suggested Action:** Add a discussion comparing your findings with those of Garfinkel et al. (2024) and Röthlisberger and Papritz (2023), highlighting how the Lagrangian and Eulerian perspectives reinforce or differ from each other, particularly regarding horizontal advection and diabatic warming.

We totally agree that comparing the Lagrangian and Eulerian perspectives with respect to heat waves would be very interesting. Many discrepancies regarding the importance of processes within heatwaves may stem from using these two different frameworks. However, we believe that we cannot fully do justice to this important comparison in this paper. To adequately address this comparison and to thoroughly take all aspects into account, we think it would require a separate paper. Therefore, we have chosen to exclusively focus on the Lagrangian perspective in this paper and not to discuss such a comparison here. However, we have added one sentence in our conclusion (Lines 505-506) to indicate that there are not only different perspectives within the Lagrangian framework, but also that an analysis in the Eulerian framework may provide yet an entirely different perspective. This reinforces the core message of our paper, namely that the characterization of heatwaves is fundamentally a matter of perspective.

- **Title and Content Alignment:**

**Comment:** The title suggests a comparison between Lagrangian and Eulerian perspectives, which the analysis does not cover explicitly.

**Suggested Action:** Consider revising the title to better reflect the paper's content or include a discussion or analysis comparing these two perspectives.

We see the reviewer's point, although the word "Eulerian" is not explicitly mentioned in the title. To avoid any misinterpretation, we decided to revise the title as suggested by the reviewer. The new title is "Two different perspectives on heat waves within the Lagrangian framework". This title

clarifies that we are exclusively addressing the Lagrangian perspective, while it also makes clear that the two perspectives we present differ.

**Conclusion**

The article is well-written and interesting, but incorporating the suggested changes will enhance its clarity, depth, and alignment with current research. Addressing these comments will strengthen the manuscript and make it more compelling for publication.

---

## Author Response (AR2)

**Response to Co-editor comment**

corncerning the manuscript

**Two different perspectives on heat waves within the Lagrangian framework**

by A. Mayer and V. Wirth,

*submitted to Weather and Climate Dynamics*

Dear Co-editor,

Thank you for your two comments, which were both very valuable. We have corrected equations (11) and (15) and added a short paragraph (Lines 329 to 333) adressing the slope of the individual curves. We believe these changes contribute to the clarity of the paper.

Best regards,

Amelie Mayer